# WrapNet: Neural Net Inference with Ultra-Low-Precision Arithmetic

**Renkun Ni**
University of Maryland
rn9zm@cs.umd.edu

**Hong-min Chu**
University of Maryland
hmchu@cs.umd.edu

**Oscar Castañeda**
ETH Zurich
caoscar@ethz.ch

**Ping-yeh Chiang**
University of Maryland
pchiang@cs.umd.edu

**Christoph Studer**
ETH Zurich
studer@ethz.ch

**Tom Goldstein**
University of Maryland
tomg@cs.umd.edu

## Abstract

Low-precision neural networks represent both weights and activations with few bits, drastically reducing the cost of multiplications. Meanwhile, these products are accumulated using high-precision (typically 32-bit) additions. Additions dominate the arithmetic complexity of inference in quantized (e.g., binary) nets, and high precision is needed to avoid overflow. To further optimize inference, we propose WrapNet, an architecture that adapts neural networks to use low-precision (8-bit) additions while achieving classification accuracy comparable to their 32-bit counterparts. We achieve resilience to low-precision accumulation by inserting a cyclic activation layer that makes results invariant to overflow. We demonstrate the efficacy of our approach using both software and hardware platforms.

## 1 Introduction

Significant progress has been made in quantizing (or even binarizing) neural networks, and numerous methods have been proposed that reduce the precision of weights, activations, and even gradients while retaining high accuracy (Courbariaux et al., 2016; Hubara et al., 2016; Li et al., 2016; Lin et al., 2017; Rastegari et al., 2016; Zhu et al., 2016; Dong et al., 2017; Zhu et al., 2018; Choi et al., 2018a; Zhou et al., 2016; Li et al., 2017; Wang et al., 2019; Jung et al., 2019; Choi et al., 2018b; Gong et al., 2019). Such quantization strategies make neural networks more hardware-friendly by leveraging fast, integer-only arithmetic, replacing multiplications with simple bit-wise operations, and reducing memory requirements and bandwidth.

Unfortunately, the gains from quantization are limited because quantized networks still require high-precision arithmetic. Even if weights and activations are represented with just one bit, deep feature computation requires the summation of hundreds or even thousands of products. Performing these summations with low-precision registers results in integer overflow, contaminating downstream computations and destroying accuracy. Moreover, as multiplication costs are slashed by quantization, high-precision accumulation starts to dominate the arithmetic cost. Indeed, our own hardware implementations show that an 8-bit $\times$ 8-bit multiplier consumes comparable power and silicon area to a 32-bit accumulator. When reducing the precision to a 3-bit $\times$ 1-bit multiplier, a 32-bit accumulator consumes more than $10\times$ higher power and area; see Section 4.5. Evidently, low-precision accumulators are the key to further accelerating quantized nets.

In custom hardware, low-precision accumulators reduce area and power requirements while boosting throughput. On general-purpose processors, where registers have fixed size, low-precision accumulators are exploited through *bit-packing*, i.e., by representing multiple low-precision integers side-by-side within a single high-precision register (Pedersoli et al., 2018; Rastegari et al., 2016; Bulat & Tzimiropoulos, 2019). Then, a single vector instruction is used to perform the same operation across all of the packed numbers. For example, a 64-bit register can be used to execute eight parallel 8-bit additions, thus increasing the throughput of software implementations. Hence, the use of low-precision accumulators is advantageous for both hardware and software implementations, provided that integer overflow does not contaminate results.

We propose WrapNet, a network architecture with extremely low-precision accumulators. WrapNet exploits the fact that integer computer arithmetic is cyclic, i.e, numbers are accumulated until they reach the maximum representable integer and then "wrap around" to the smallest representable integer. To deal with such integer overflows, we place a differentiable cyclic (periodic) activation function immediately after the convolution (or linear) operation, with period equal to the difference between the maximum and minimum representable integer. This strategy makes neural networks resilient to overflow as the activations of neurons are unaffected by overflows during convolution.

We explore several directions with WrapNet. On the software side, we consider the use of bit-packing for processors with or without dedicated vector instructions. In the absence of vector instructions, overflows in one packed integer may produce a carry bit that contaminates its neighboring value. We propose training regularizers that minimize the effects of such contamination artifacts, resulting in networks that leverage bit-packed computation with very little impact on final accuracy. For processors with vector instructions, we modify the Gemmlowp library (Jacob et al., 2016) to operate with 8-bit accumulators. Our implementation achieves up to $2.4\times$ speed-up compared to a 32-bit accumulator implementation, even when lacking specialized instructions for 8-bit multiply-accumulate. We also demonstrate the efficacy of WrapNet in terms of cycle time, area, and energy efficiency when considering custom hardware designs in a commercial 28 nm CMOS technology.

## 2 RELATED WORK AND BACKGROUND

### 2.1 NETWORK QUANTIZATION

Network quantization aims at accelerating inference by using low-precision arithmetic. In its most extreme form, weights and activations are both quantized using binary or ternary quantizers. The binary quantizer $Q_b$ corresponds to the sign function, whereas the ternary quantizer $Q_t$ maps some values to zero. Multiplications in binarized or ternarized networks (Hubara et al., 2016; Courbariaux et al., 2015; Lin et al., 2017; Rastegari et al., 2016; Zhu et al., 2016) can be implemented using bitwise logic, leading to impressive acceleration. However, training such networks is challenging since fewer than 2 bits are used to represent activations and weights, resulting in a dramatic impact on accuracy compared to full-precision models.

Binary and ternary networks are generalized to higher precision via uniform quantization, which has been shown to result in efficient hardware (Jacob et al., 2018). The multi-bit uniform quantizer $Q_u$ is given by: $Q_u(x) = \text{round}(x/\Delta_x)\Delta_x$, where $\Delta_x$ denotes the quantization step-size. The output of the quantizer is a floating-point number $x$ that can be expressed as $x = \Delta_x x_q$, where $x_q$ is the fixed-point representation of $x$. The fixed-point number $x_q$ has a "precision" or "bitwidth," which is the number of bits used to represent it. Note that the range of floating-point numbers representable by the uniform quantizer $Q_u$ depends on both the quantization step-size $\Delta_x$ and the quantization precision. Nonetheless, the number of different values that can be represented by the same quantizer depends only on the precision.

Applying uniform quantization to both weights $w = \Delta_w w_q$ and activations $x = \Delta_x x_q$ simplifies computations, as an inner-product simply becomes

$$z = \sum_i w_i x_i = \sum_i (\Delta_w (w_q)_i)(\Delta_x (x_q)_i) = (\Delta_w \Delta_x) \sum_i (w_q)_i (x_q)_i = \Delta_z z_q. \qquad (1)$$

The key advantage of uniform quantization is that the core computation $\sum_i (w_q)_i (x_q)_i$ can be carried out using fixed-point (i.e., integer) arithmetic only. Results in (Gong et al., 2019; Choi et al., 2018b; Jung et al., 2019; Wang et al., 2019; Mishra et al., 2017; Mishra & Marr, 2017) have shown that high classification accuracy is attainable with low-bitwidth uniform quantization, such as 2 or 3 bits. Although $(w_q)_i, (x_q)_i$, and their product may have extremely low-precision, the accumulated result $z_q$ of many of these products has very high dynamic range. As a result, high-precision accumulators are typically required to avoid overflows, which is the bottleneck for further arithmetic speedups.

### 2.2 LOW-PRECISION ACCUMULATION

Several approaches have been proposed that use accumulators with fewer bits to obtain speed-ups. For example, reference (Khudia et al., 2021) splits the weights into two separate matrices, one with

Table 1: Average overflow rate (in 8 bits) of each layer for a low-precision network and corresponding test accuracy using either 32-bit or 8-bit accumulators during inference on CIFAR10.

| Bit (A/W) | Overflow rate (8-bit) | Accuracy (32-bit) | Accuracy (8-bit) |
|---|---|---|---|
| full precision | – | 92.45% | – |
| 3/1 | 10.84% | 91.08% | 10.06% |
| 2/1 | 1.72% | 88.46% | 44.04% |

small- and another with large-magnitude entries. If the latter matrix is sparse, acceleration is attained as most computations rely on fast, low-precision operations. However, to significantly reduce the accumulator's precision, one would need to severely decrease the magnitude of the entries of the first matrix, which would, in turn, prevent the second matrix from being sufficiently sparse to achieve acceleration. Recently, (de Bruin et al., 2020) proposed using layer-dependent quantization parameters to avoid overflowing accumulators with fixed precision. Fine-tuning is then used to improve performance. However, if the accumulator precision is too low (e.g., 8 bits or less), the optimized precision of activations and weights is too coarse to attain satisfactory performance. Another line of work (Sakr et al., 2019; Micikevicius et al., 2017; Wang et al., 2018) uses 16-bit floating-point accumulators for training and inference—such approaches typically require higher complexity than methods based on fixed-point arithmetic.

## 2.3 THE IMPACT OF INTEGER OVERFLOW

Overflow is a major problem, especially in highly quantized networks. Table 1 demonstrates that overflows occur in around 11% of the neurons in a network with 3-bit activations (A) and binary weights (W) that is using 8-bit accumulators for inference after being trained on CIFAR-10 with standard precision. Clearly, overflow has a significant negative impact on accuracy. Table 1 shows that if we use an 8-bit (instead of a 32-bit) accumulator, then the accuracy of a binary-weight network with 2-bit activations drops by more than 40%, even when only 1.72% neurons overflow. If we repeat the experiment with 3-bit activations and binary weights, the accuracy is only marginally better than a random guess. Therefore, existing methods try to *avoid* integer overflow by using accumulators with relatively high precision, and pay a correspondingly high price when doing arithmetic.

## 3 WRAPNET: DEALING WITH INTEGER OVERFLOWS

We now introduce WrapNet, which includes a cyclic activation function and an overflow penalty, enabling neural networks to use low-precision accumulators. We also present a modified quantization step-size selection strategy for activations, which retains high classification accuracy. Finally, we show how further speed-ups can be achieved on processors with or without specialized vector instructions.

We propose training a network with layers that emulate integer overflows on the fixed-point pre-activations $z_q$ to maintain high accuracy. However, directly training a quantized network with an overflowing accumulator diverges (see Table 2) due to the discontinuity of the modulo operation. To facilitate training, we insert a cyclic "smooth modulo" activation immediately after every linear/convolutional layer, which not only captures the wrap-around behavior of overflows, but also ensures that the activation is continuous everywhere. The proposed smooth modulo activation $c$ is a composite function of a modulo function $m$ and a basis function $f$ that ensures continuity. Specifically, given a $b$-bit accumulator, our smooth-modulo $c$ for fixed-point inputs is as follows:

$$f(m) = \begin{cases} m, & \text{for } -\frac{k}{k+1}2^{b-1} \leq m \leq \frac{k}{k+1}2^{b-1} \\ -k2^{b-1} - km, & \text{for } m < -\frac{k}{k+1}2^{b-1} \\ k2^{b-1} - km, & \text{for } m > \frac{k}{k+1}2^{b-1} \end{cases}$$
$$c(z_q) = f(\text{mod}(z_q + 2^{b-1}, 2^b) - 2^{b-1}),$$

where $k$ is a hyper-parameter that controls the slope of the transition. Note that we apply constant shifts to keep the input of $f$ in $[-2^{b-1}, 2^{b-1})$. Figure 1a illustrates the smooth modulo function with

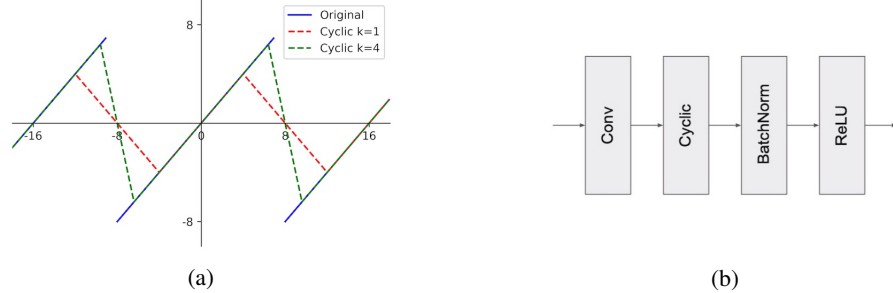

(a)                                                         (b)

Figure 1: (a) Example of the proposed cyclic activation with different slopes $k$ and the original modulo operator for a 4-bit accumulator. (b) Convolutional block with proposed cyclic activation.

two different slopes $k = 1, 4$. As $k$ increases, the cyclic activation becomes more similar to the modulo operator and has a greater range, but the transition becomes more abrupt. Since our cyclic activation is continuous and differentiable almost everywhere, standard gradient-based learning can be applied easily. A convolutional block with cyclic activation layer is shown in Figure 1b. After the convolution result goes into the cyclic activation, the result is multiplied by $\Delta_z$ to compute a floating-point number, which is then processed through BatchNorm and ReLU. A fixed per-layer quantization step-size is then used to convert the floating-point output of the ReLU into a fixed-point input for the next layer. We detail the procedure to find this step-size in Section 3.2.

### 3.1 OVERFLOW PENALTY

An alternative way to adapt quantized networks to low-precision accumulators is to directly reduce the amount of overflows. To achieve this, we propose a regularizer which penalizes outputs that exceed the bitwidth of the accumulation register. Concretely, for a $b$-bit accumulator, we define an overflow penalty for the $l$-th layer of the network as follows: $R_l^o = (1/N) \sum_i \max\{|z_q^i| - 2^{b-1}, 0\}$. Here, $z_q^i$ is the fixed-point result in (1) for the $i$-th neuron of the $l$-th layer, and $N$ is the total number of neurons in the $l$-th layer. The overflow penalty is imposed after every quantized linear layer and before the cyclic activation. All these penalties are combined into one regularizer $R^o = \sum_l R_l^o$.

### 3.2 SELECTION OF ACTIVATION QUANTIZATION STEP-SIZE

To keep multiplication simple, the floating-point output of ReLU must be quantized before it is fed into the following layer. However, as shown in Table 1, a significant number of overflows occur even with 3-bit activations. From our experiments (see Table 3), we have observed that if overflow occurs too frequently (i.e., on more than 10% of the neurons), then WrapNet starts to suffer significant accuracy degradation. However, if we reduce the activation precision so that no overflows happen at all, several layers will have 1-bit activations (see Table 3), thereby increasing quantization errors and degrading accuracy. To balance accumulation and quantization errors, we adjust the quantization step-size $\Delta_x$ of each layer based on the overflow rate, i.e., the percentage $p\%$ of neurons that overflow in the network. If the overflow rate $p\%$ is too large, then we increase $\Delta_x$ to reduce the overflow rate $p\%$. The selected quantization step-size is then fixed for further fine-tuning.

### 3.3 ADAPTING TO BIT-PACKING

Most modern processors provide vector instructions that enable parallel operation on multiple 8-bit numbers. For instance, the AVX2 (NEON) instruction set on x86 (ARM) processors provides parallel processing with 32 (16) 8-bit numbers. Vector instructions provide a clean implementation of bit-packing, which WrapNet can leverage to attain significant speed-ups. While some embedded processors and legacy chips do not provide vector instructions, bit-packing can still be applied. Without vector instructions for multiplication, binary/ternary weights must be used to replace multiplication with bit-wise logic (Bulat & Tzimiropoulos, 2019; Pedersoli et al., 2018). Furthermore, bit-packing of additions is more delicate: Each integer overflow not only results in wrap-around behavior, but also generates a carry bit that contaminates the adjacent number—specialized vector

instructions avoid such contamination. We propose the following strategies to minimize the impact of carry propagation.

**Reducing variance in the number of carries.** The number of carries generated during a convolution operation can be large. Nevertheless, if we can keep the number of carries approximately the same for all the neurons among a batch of images, the estimated number of carries can be subtracted from the result to correct the outputs of a bit-packed convolution operation. To achieve this, during training, we calculate the number of carries for each neuron and impose a regularizer, $R^c$, to keep the variance of the number of carries small. The detailed formulation of $R^c$ can be found in Appendix A.1. **Using a buffer bit.** Alternatively, since each addition can generate at most one carry bit, we can place a buffer bit between every low-bit number in the bit-packing. For example, instead of packing eight 8-bit representations into a 64-bit number, we pack eight 7-bit numbers with one buffer bit between each of them. These buffer bits absorb the carry bits, and are cleared using bit-wise logic after each addition. Buffering makes representations 1-bit smaller, which potentially degrades accuracy. **A hybrid approach.** To get the benefits from both strategies, we use a variance penalty on layers that have small standard deviation to begin with, and equip the remaining layers with a buffer bit.

## 4 EXPERIMENTS

We compare the accuracy and efficiency of WrapNet to networks with full-precision accumulators using the CIFAR-10 and ImageNet datasets. Most experiments use binary or ternary weights for WrapNet as AVX2 lacks 8-bit multiplication instructions, but supports 8-bit additions and logic operations needed for binary/ternary convolutions.

### 4.1 TRAINING PIPELINE

We first pre-train a network with quantized weights and no cyclic layers, while keeping full-precision activations. Then, we select the quantization step-sizes of the activations (see Section 3.2) such that each layer has an overflow rate of around $p\%$ (a hyper-parameter) with respect to the desired accumulator bitwidth. Given the selected quantization step-size for each layer and the pre-trained network, we insert our proposed cyclic activation layer. We then warm-up our WrapNet by fine-tuning with full-precision activation for several epochs. Finally we further fine-tune the network with both activations and weights quantized. Both overflow and carry variance regularizers are only applied in the final fine-tuning step, except when training ResNet for ImageNet, where the regularizers are also included during warm-up.

### 4.2 ADAPTING TO LOW-PRECISION ACCUMULATORS

We conduct ablation studies on the following factors: the type of cyclic function, the initial overflow rate for quantization step-size and precision selection, and the coefficient of the overflow penalty regularizer. These experiments are conducted on VGG-7 (Li et al., 2016), which is commonly used in the quantization literature for CIFAR-10. We binarize the weights as in (Rastegari et al., 2016), and we train WrapNet to adapt to an 8-bit accumulator. As our default setting, we use $k = 2$ as the transition slope, $p = 5\%$ as the initial overflow rate, and $0$ as the coefficient of the regularizer.

**Cyclic activation function.** We compare the performance of various transition slopes $k$ of our cyclic function $c$ in Table 2, and we achieve the best performance when $k = 2$. If $k$ is too small, then the accuracy decreases due to a narrower effective bitwidth (only half of the bitwidth is used when $k = 1$). Meanwhile, the abrupt transition for large $k$ hurts the performance as well. In the extreme case where the cyclic function degenerates to modulo ($k \to \infty$), WrapNet diverges to random guessing, which highlights the importance of training with a "smooth" cyclic non-linearity to assimilate integer overflow. We also find that placing a ReLU after batch norm yields the best performance, even though the cyclic function is already non linear. More experimental results can be found in Appendix B.1.

**Quantization step-size.** As described in Section 3.2, the quantization step-sizes are selected to balance the rounding error of the activations and accumulation errors due to overflow. We compare the classification performance when we choose different step-sizes to control the overflow rate as in

Table 2: Results for different transition slopes for cyclic function; $*$ denotes divergence.

| $k$ | 1 | 2 | 4 | 10 | $\infty$ |
|---|---|---|---|---|---|
| Accuracy | 90.24% | 90.52% | 90.25% | 89.16% | $*$ |

Table 3: Results for different quantization step-sizes based on overflow rate $p(\%)$. $*$ denotes divergence.

Table 4: Results for fine-tuning with the overflow penalty ($R^o$).

| $p$ | Bits | Accuracy | $p$ | Bits | Accuracy |
|---|---|---|---|---|---|
| 0 | 1 | 90.07% | 20 | 4 | 88.25% |
| 2 | 3 | 90.51% | 30 | 5 | 85.30% |
| 5 | 3 | 90.52% | 40 | 5 | 36.11% |
| 10 | 4 | 89.92% | 50 | 5 | $*$ |

| $R^o$ | $p\%$ | Accuracy | Difference |
|---|---|---|---|
| 0 | 20 | 88.25% | – |
| 0 | 5 | 90.52% | 2.27% |
| 0.01 | 20 | 90.05% | – |
| 0.01 | 5 | 90.81% | 0.76% |

Table 3. If the initial overflow rate is large, then the quantization step-size will be finer, but training is less stable. We obtain the best performance when the initial overflow rate is around 5%. The median bitwidths of the activations across layers are also reported in Table 3. Note that if we want to suppress all overflows, we can only use 1-bit activations. We also observe that WrapNet can attain reasonable accuracy (85%) even with a large overflow rate (around 30%), which demonstrates that our proposed cyclic activations provides resilience against integer overflows.

**Overflow penalty.** The overflow penalty regularizer improves stability to step-size selection. More specifically, in Table 4, the difference in accuracy between two step-size selections decreases from 2.27% to 0.76% after adding the regularizer. The overflow penalty also complements our cyclic activation, as we achieve the best performance when using both of them together during the fine-tuning stage. Moreover, in Appendix B.2, we compare our results to fine-tuning the pre-trained network using the overflow regularizer only. In the absence of a cyclic layer, neural networks still suffer from low accuracy (as in Section 2.3) unless a very strong penalty is imposed.

## 4.3 ADAPTING TO BIT-PACKING

We now show the efficacy of WrapNet for bit-packing without vector operations. We use the same architecture, binary weights, 8-bit accumulators, and hyper-parameters as in Section 4.2. The training details can be found in Appendix A.2. We consider CIFAR-10, and we compare with the best result of WrapNet from the previous section as a baseline. Without specific vector instructions, accuracy degenerates to a random guess because of undesired carry contamination during inference.

Surprisingly, with the carry variance regularizer, WrapNet works well even with abundant carry contamination during inference (for each neuron, 384 on average over all the dataset). The regularizer drops the standard deviation of the per-neuron carry contamination by 90%. When we use the hybrid approach, the accuracy is further improved (89.43%) and close to the best result (90.81%) we can achieve with vector instructions that do not propagate carries across different numbers (see Table 5).

Table 5: Results for adaptation to bit-packing with 8-bit accumulator. (v) denotes no carry contamination as in a vector instruction; (c) denotes carry propagation between different numbers.

| Method | Accuracy (v) | Accuracy (c) | Carry | Carry Std |
|---|---|---|---|---|
| Baseline | 90.81% | 10.03% | 254.91 | 159.55 |
| Buffer Bit | – | 88.22% | – | – |
| $R^c$ | – | 87.86% | 384.42 | 17.91 |
| Hybrid | – | 89.43% | 482.4 | 16.18 |

## 4.4 BENCHMARK RESULTS

In this section, we compare our WrapNet when there is no carry contamination, with the following 32-bit accumulator baselines: a full-precision network (FP), a network trained with binary/ternary weights but with full-precision activations (BWN/TWN), and a network where both weights and activations are quantized to the same precision as our WrapNet (BWN/TWN-QA). We benchmark our results on both CIFAR-10 and ImageNet. We use VGG7 and ResNet20 for our CIFAR-10 experiments, and we use AlexNet (Krizhevsky et al., 2012; Simon et al., 2016), ResNet18 and ResNet50 (He et al., 2016) for our ImageNet experiments. Details of training can be found in Appendix B.3.

For CIFAR-10, even with an 8-bit accumulator, our results are comparable to both BWN and TWN. When adapting to a 12-bit accumulator, we further achieve performance on-par with TWN and better than BWN (see Table 6). For ImageNet, our WrapNet can achieve accuracy as good as BWN when adapting to a 12-bit accumulator where we can use binary weights and roughly 7-bit quantized activations. However, in the extreme low-precision case (8-bit), the accuracy of our binary WrapNet drops around 8% due to the limited bitwidth we can use for activations. As reported in Table 6, the median activation bitwidth is roughly 3-bit, and for some layers in AlexNet, we can only use 1-bit activations. Despite the gap from BWN, we observe that our model can achieve comparable performance as BWN-QA where the same precision is used for activations. When using ternary weights and an 8-bit accumulator, our WrapNet only drops by 3% and 2% from TWN for ResNet18 and ResNet50, respectively. In addition, in the case of adapting to a 12-bit accumulator, our ternary WrapNet with roughly 7-bit activations is even slightly better than TWN for ResNet50. Note that, without cyclic activation function, all the results for networks using 8-bit accumulator are as poor as random guessing which is consistent with Table 1.

Table 6: Top-1 test accuracy for both CIFAR-10 and ImageNet with different architectures. Here, "Acc" represents accumulator, and "QA" represents quantized activation.

|  | Bits | | | CIFAR-10 | | ImageNet | | |
|---|---|---|---|---|---|---|---|---|
|  | Activation | Weight | Acc | VGG7 | ResNet20 | AlexNet | ResNet18 | ResNet50 |
| FP | 32 | 32 | 32 | 92.45% | 91.78% | 60.61% | 69.59% | 76.15% |
| BWN | 32 | 1 | 32 | 91.55% | 90.03% | 56.56% | 63.55% | 72.88% |
| BWN-QA | $\sim 3$ | 1 | 32 | 91.30% | 89.86% | 46.30% | 57.54% | 66.85% |
| WrapNet | $\sim 3$ | 1 | 8 | 90.81% | 89.78% | 44.88% | 55.60% | 64.30% |
| WrapNet | $\sim 7$ | 1 | 12 | 91.59% | 90.17% | 56.62% | 63.11% | 72.37% |
| TWN | 32 | 2 | 32 | 91.56% | 90.36% | 57.57% | 65.70% | 73.31% |
| TWN-QA | $\sim 4$ | 2 | 32 | 91.49% | 90.12% | 55.84% | 63.67% | 72.50% |
| WrapNet | $\sim 4$ | 2 | 8 | 91.14% | 89.56% | 52.24% | 62.13% | 71.62% |
| WrapNet | $\sim 7$ | 2 | 12 | 91.53% | 90.88% | 57.60% | 63.84% | 73.93% |

## 4.5 EFFICIENCY ANALYSIS

We conduct an efficiency analysis of parallelization by bit-packing, both with and without vector operations, on an Intel i7-7700HQ CPU operating at 2.80 GHz. We also conduct a detailed study of improvements that can be obtained using custom hardware.

**AVX2 instruction efficiency analysis.** We study the empirical efficiency of WrapNet when vector operations are available. We extended Gemmlowp (Jacob et al., 2016) to implement matrix multiplications using 8-bit accumulators with AVX2 instructions. To demonstrate the efficiency of low-precision accumulators, we compare our implementation with the AVX2 version of Gemmlowp, which uses 32-bit accumulators. We report the execution speed of both on various convolution kernels of ResNet18 in Table 7. From Table 7 we observe significant speed-ups ranging from $2\times$ to $2.4\times$ among different blocks. Besides, we compare the entire inference time (ms) of ResNet18 for WrapNet (234.74) with a 32b-accumulator quantized network (312.42), which gains 33% speed-up. The result provides solid evidence for the efficiency advantage of using low-precision accumulators. We remark that in average, the time cost for cyclic activation is only around 10% of the time cost

Table 7: Time cost (ms) for typical $3 \times 3$ convolution kernels in ResNet using different accumulator bitwidths.

| Input size | Output | 8-bit | 32-bit |
|---|---|---|---|
| 64x56x56 | 64 | **3.467** | 8.339 |
| 128x28x28 | 128 | **2.956** | 6.785 |
| 256x14x14 | 256 | **2.499** | 5.498 |
| 512x7x7 | 512 | **2.710** | 5.520 |

Table 8: Time cost (ms) for $3 \times 3$ convolution kernels in ResNet with no vector instructions using bit packing.

| Input size | Output | bit packing | naïve |
|---|---|---|---|
| 64x56x56 | 64 | **29.80** | 83.705 |
| 128x28x28 | 128 | **23.86** | 80.557 |
| 256x14x14 | 256 | **21.71** | 86.753 |
| 512x7x7 | 512 | **20.41** | 87.671 |

for the GEMM kernel. We also remark that AVX2 lacks a single instruction that performs both multiplication and accumulation for 8-bit data, but it does have such instruction for 32-bit data. Thus, further acceleration can be achieved on systems like ARM where such combined instructions for 8-bit data are available.

**Bit-packing results without vector operations.** We implement a naïve for-loop based matrix multiplication, which uses buffer bit and logical operations introduced in Section 3.3 to form the baseline. We then pack four 8-bit integers into 32 bits, and report the execution speed of both implementations on various convolution kernels of ResNet18 in Table 8. The results show significant speed-ups ranging from $2.8\times$ to $4.3\times$. Such observations demonstrate our proposed approach to handle extra carry bits makes bit-packing viable and efficient, even when vector instructions are not available.

**Hardware analysis.** To illustrate the potential benefits of WrapNet for custom hardware accelerators, we have implemented a multiply-accumulate (MAC) unit in a commercial 28nm CMOS technology. The MAC unit consists of (i) a multiplier with an output register, (ii) an accumulator with its corresponding register, and (iii) auxiliary circuitry. Please refer to Appendix C for the details. We have considered 8-bit $\times$ 8-bit and 3-bit $\times$ 1-bit multipliers, as well as 32-bit and 8-bit accumulators, where the latter option is enabled by our WrapNet approach and its cyclic activation function. We consider a slope $k = 2$ for the cyclic activation. Figure 2 shows our post-layout results.

Figure 2a shows that reducing the multiplier bitwidth decreases the cycle time by 7%; reducing the accumulator precision from 32-bit to 8-bit further the cycle time by 16%. Figures 2b and 2c highlight the importance of reducing the accumulator's precision. When using an 8-bit $\times$ 8-bit multiplier, the 32-bit accumulator already constitutes more than 40% of the area and energy of a MAC unit. Once the multiplier's precision reduces, the accumulator dominates area- and energy-efficiency. Thanks to WrapNet, we can reduce the accumulator precision from 32-bit to 8-bit, thus reducing the accumulator's area- and energy-efficiency by more than $5\times$ and $4\times$, respectively. WrapNet requires the implementation of the cyclic activation, which has an area- and energy-efficiency comparable (although lower) to that of the accumulator. In spite of this overhead, WrapNet is still able to reduce the total MAC unit's area- and energy-efficiency by up to $3\times$ and $2\times$, respectively. While our hardware implementation only uses one adder per inner-product, we note that WrapNet can also be applied to spatial architectures, such as systolic arrays, which use several adders per inner-product. For such spatial architectures, WrapNet avoids an increase in the adders' bitwidth, normalizing all

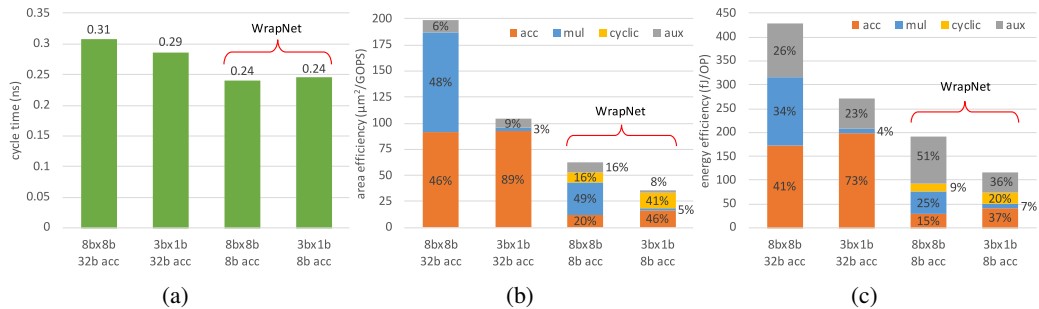

Figure 2: (a) Cycle time, (b) area and (c) energy efficiency for different MAC units implemented in 28nm CMOS. We consider 8-bit $\times$ 8-bit or 3-bit $\times$ 1-bit multipliers with 32-bit or 8-bit accumulators.

adders to the same low bitwidth. Moreover, the use of several adders per inner-product amortizes the overhead from the cyclic activation, of which only one is needed per inner-product. Finally, we note that this analysis only considers the computation part of a hardware accelerator as this is where WrapNet has a significant impact—the memory sub-system will remain virtually the same, as existing methods already quantize the output activations to low-bit before storing them in memory.

## 5 CONCLUSION

We have proposed WrapNet, a novel method to render neural networks resilient to integer overflow, which enables the use of low-precision accumulators. We have demonstrated the effectiveness of our adaptation on both CIFAR-10 and ImageNet. In addition, our custom GEMM kernel achieves $2.4\times$ acceleration over its standard library version, and our hardware exploration shows significant improvements in area- and energy-efficiency. Our hope is that hardware-aware architectures will enable deep learning applications on a wide range of platforms and mobile devices. Furthermore, with future innovations in GPU and data center technologies, we hope that WrapNet can provide further speed-ups by enabling inference using quarter-precision—a step forward in terms of performance from the currently available half-precision standard available on emerging GPUs.

## ACKNOWLEDGEMENT

The university of Maryland team was supported by the ONR MURI program, AFOSR MURI program, and the National Science Foundation DMS division. Addition support was provided by DARPA GARD, DARPA QED4RML, and DARPA YFA.

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

## A    DETAILS OF CARRY VARIANCE REDUCTION REGULARIZER

### A.1    CARRY VARIANCE CALCULATION

With two's complement representations for signed integers, a carry bit is generated in the following three cases: (i) addition of two negative numbers, (ii) addition of two positive numbers whose result exceeds the representation range, thus provoking integer overflow, and (iii) addition of a positive and a negative number whose result is a positive number. Dealing with these cases individually is complicated, but the calculation can be simplified by first reinterpreting the two's complement representation as an unsigned integer. Carry bits resulting from accumulation of unsigned integers is easier to calculate as they can only happen in case (ii) as described above.

Since we only consider binary/ternary weights for bit-packing, carry bits can only be generated during accumulation, and not by multiplication. To produce a single output from a convolution, we must perform the accumulation $\sum_{i=1}^{L} \mathbf{v}_i$ of all entries of the vector $\mathbf{v} \in \mathbb{R}^L$.. This is done by batching computations inside a $b$-bit register as follows. First, we bit-pack groups of numbers $\mathbf{v}_i$ into several high-resolution registers. For example, let us consider the use of 32-bit registers to pack four $b = 8$-bit numbers; then, we need to use $\lceil L/4 \rceil$ 32-bit registers to represent all $L$ entries of $\mathbf{v}$. In the absence of vector instructions, the addition of these high-resolution registers will generate carry bits that will contaminate the adjacent bit-packed numbers. After all $\lceil L/4 \rceil$ additions take place, we add the 4 bit-packed numbers together to get a final result.

When one output feature is calculated by bit-backing as described above, the effect of carry bits is easy to simulate; accumulations can be done without accounting for carry bits during convolution, and then the carry bits can be added into the the final result after convolution takes place. If the total number of carries is large, this final correction can in turn produce new carry bits. Hence, we use Algorithm 1 to compute the total number of carry bits that are generated in an accumulation. The first equation simply reinterprets the signed representation to its unsigned counterpart $u$. Then, we compute the amount of carry bits $c_i$, as well as the result $r_i$ remaining within the $b$-bit accumulator. Due to carry contamination, the carry bits $c_i$ will be added to the result $r_i$, which may generate new carry bits $c_{i+1}$. We keep on adding the new carry bits to the accumulator until no new carry bits are generated. Note that, in real hardware at inference time, the most signifiant carry bit produced inside a register will be thrown away. For simplicity, our simulations during training accumulate all carry bits, including the most significant. We find that dropping the most significant carry during inference does not significantly impact testing.

---

**Algorithm 1:** Carry Amount Calculation

initialization $\mathbf{v}, b$;
$u = \sum_i \left( (\text{sign}(\mathbf{v}_i) + 1)/2 \right) \mathbf{v}_i + \left( (-\text{sign}(\mathbf{v}_i) + 1)/2 \right) \left( \mathbf{v}_i + 2^b \right)$;
$c_i = u, r_i = 0, c = 0$;
**while** $c_i \neq 0$ **do**
  $\quad c_{i+1} = \left\lfloor (c_i + r_i)/2^b \right\rfloor$;
  $\quad r_{i+1} = (c_i + r_i) \bmod 2^b$;
  $\quad c = c + c_{i+1}$;
  $\quad c_i = c_{i+1}, r_i = r_{i+1}$
**end**
**return** $c$

---

Given the number of carry bits calculated during the inner product, the variance of the carry among a batch $(b_s)$ of images is calculated as follows:

$$\mathrm{m}_{b_s}(n^{i,l}) = \frac{1}{b_s} \left( \sum_{k=1}^{b_s} n_k^{i,l} \right), \tag{2}$$

$$\mathrm{var}_{b_s}(n^{i,l}) = \frac{1}{b_s} \left( \sum_{k=1}^{b_s} \left( n_k^{i,l} - \mathrm{m}_{b_s}(n^{i,l}) \right)^2 \right), \tag{3}$$

where $n^{i,l}$ is the carry bit for the $i$-th neuron in $l$-th layer (assuming all the feature maps are vectorized). The estimated mean among all the images are learned by a moving average based on the

mean of batches equation 2. However, the sign and rounding function may have zero gradient almost everywhere. To make all the operations differentiable, we replace the sign function with a tanh function and we use a straight through estimator for rounding during the backward pass (gradient is identity). Then finally, our regularizer $R^c$ will be the mean variance among all the neurons.

## A.2 TRAINING WITH CARRY VARIANCE REDUCTION REGULARIZER

Due to the large amount and high variance of carry-bit occurrences, it is hard to fine-tune our Wrap-Net even when using the carry variance reduction regularizer. The generated carry bits will be accumulated, which increases the overflow rate dramatically. In addition, the accumulation error will contaminate downstream computations and destroy accuracy. As a result, we fine-tune WrapNet with simulated carry bits layer by layer, starting from the layer which has the least carry variance. For the hybrid approach, we stop simulating the carry bit when we notice a significant accuracy drop; the remaining layers are trained using a buffer bit instead.

## B EXPERIMENTAL DETAILS

### B.1 MORE CYCLIC FUNCTIONS

We compare two more "smooth" cyclic functions with our proposed cyclic activation function in Section 3. Specifically, we consider a cyclic absolute value function, and a ReLU-like function with transition slope $k$ as alternative cyclic activations. Figure 3 illustrates the compared functions. We compare the results with and without a ReLU activation after batch normalization as well. Table 9 shows that retaining the ReLU activation after the batch normalization layer always achieves a better result, and that our proposed cyclic activation outperforms the other two choices.

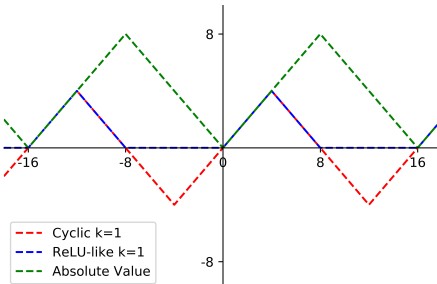

Figure 3: Example of the compared cyclic functions for a 4-bit accumulator.

Table 9: Results for different types of cyclic activation

| Cyclic Function | ReLU | slope $k$ | Accuracy(%) |
|---|---|---|---|
| Proposed | √ | 2 | **90.52** |
| Proposed | | 2 | 89.28 |
| ReLU-like | √ | 1 | 90.25 |
| ReLU-like | √ | 2 | 90.31 |
| ReLU-like | √ | 3 | 90.15 |
| ReLU-like | | 1 | 88.62 |
| ReLU-like | | 2 | 89.01 |
| ReLU-like | | 3 | 88.53 |
| Absolute | √ | – | 90.17 |
| Absolute | | – | 89.19 |

### B.2 FULL OVERFLOW PENALTY RESULTS

Table 10 shows the results for fine-tuning our WrapNet with different coefficients for the overflow penalty. When applying the overflow penalty, the overflow rate decreases and we can achieve a

higher accuracy. In addition, when we apply the regularizer to a network with low-resolution accumulators that does not use our cyclic activation, the network still suffers from performance degradation unless a large coefficient is used. However, a strong penalty kills almost all of the overflow, which may limit the performance of a deep neural network.

Table 10: Comparison for fine-tuning network without cyclic activation and our WrapNet, with overflow penalty $R^o$.

| Cyclic | $R^o$ | Overflow rate (%) | Accuracy(%) |
|:------:|:-----:|:-----------------:|:-----------:|
| ✓ | 0 | 6.29 | 90.52 |
| ✓ | 0.001 | 1.88 | 90.33 |
| ✓ | 0.01 | 1.24 | **90.81** |
| ✓ | 0.1 | 1.04 | 89.52 |
| | 0.01 | 5.91 | 64.69 |
| | 0.1 | 0.35 | 88.94 |
| | 1 | 0.06 | 90.26 |
| | 2 | 0.03 | 90.20 |

### B.3 TRAINING DETAILS FOR BENCHMARK RESULTS

For fair comparison, all our baselines (BWN/TWN, BWN-/TWN-QA) are fine-tuned from a pretrained full-precision network. We leave the first and last layer at full-precision as in (Rastegari et al., 2016; Zhou et al., 2016). To obtain the benchmark results of our WrapNet, we follow a training pipeline, where we first warm-up our WrapNet with full-precision activations, and then we fine-tune the network for quantized activations. We set the transition slope $k = 2$, and the initial overflow rate $p = 5\%$. The overflow penalty coefficients for CIFAR-10 and ImageNet are 0.01 and 0.001, respectively.

For the CIFAR-10 results, we use ADAM as our optimizer with an initial learning rate of 0.001. For both warm-up and fine-tuning stages, we run 200 epochs, and the learning rate is divided by 10 every 60 epochs. For all the ImageNet results, we use SGD with momentum 0.9, weight decay $1 \times 10^{-4}$ as our optimizer. We run 60 epochs for both warm-up and fine-tuning stages, where the initial learning rate is 0.01, which is divided by 10 at (20, 40, 50) epochs. We note that, due to depth of ResNet, we select a fixed quantization step-size for all the layers, where the average initial overflow rate is around 5%. As a result, the overflow penalty is also imposed during the warm-up stage for ResNet experiments.

## C HARDWARE ANALYSIS

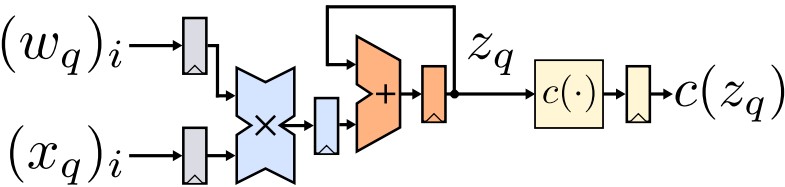

Figure 4: Multiply-accumulate (MAC) unit, together with cyclic activation function $c(\cdot)$, implemented for hardware analysis.

Figure 4 shows the multiply-accumulate (MAC) unit implemented in TSMC 28nm CMOS. The MAC unit multiplies two scalars and accumulates these products using an adder. To perform this functionality, the MAC unit is composed of multiplication, accumulation, and auxiliary circuitry, colored in Figure 4 with blue, orange, and gray, respectively. Clock distribution circuitry is not shown, but is included in our results as part of the auxiliary circuitry. Furthermore, we have implemented the cyclic activation function in hardware, colored in Figure 4 with yellow, which is only

used together low-bitwidth accumulators. To achieve lower cycle times (i.e., a faster operation frequencies), as well as to separate the multiplier's and accumulator's critical paths, we introduced a pipeline register between the multiplier and accumulator. For our implementation results, we consider this pipeline register as part of the multiplication circuitry.

We implemented the circuit in Figure 4 using different bitwidths for the multiplier (8-bit×8-bit or 3-bit×1-bit) and the accumulator (32-bit or 8-bit). When using the 8-bit×8-bit multiplier with the 32-bit accumulator, we use 16 bits for the multiplier's output register to represent all possible products. When using the 8-bit×8-bit multiplier with the 8-bit accumulator, we use 8 bits for the multiplier's output register, since the accumulator does not support larger bitwidths. When using the 3-bit×1-bit multiplier, we use 4 bits for the multiplier's output register, regardless of the accumulator's bitwidth. The cyclic activation is only implemented when using the 8-bit accumulator, for both multiplier's bitwidth. We implemented the cyclic activation for slopes of $k = 2$ and $k = 4$.

The four different MAC units were synthesized using Synopsys Design Compiler (DC), and automatically placed-and-routed using Cadence Innovus. Power analysis was done using Cadence Innovus with stimuli-based post-layout simulations at 0.9V and 25°C in the typical-typical corner. For the stimuli, we used weights and activations extracted from a layer of the ResNet-18 network. Tables 11, 12, and 13 show the implementation results from Figure 2 in tabular form. Note that throughput is computed as 2/cycle time, as the MAC unit completes two operations (multiplication and accumulation) in a single clock cycle. However, in Figure 2, we decided to report cycle time so that, for all metrics presented (cycle time, area- and energy-efficiency), a lower value corresponds to a better performance. Note that circuits with a higher throughput (which corresponds, in this case, to a lower cycle time) often result in higher area and power consumption. As a matter of fact, dynamic power consumption is directly proportional to operation frequency (i.e., 1/cycle time). Thus, to perform a fair comparison, we have normalized the area and power reported in Table 11 by the throughput achieved, resulting in the area- and energy-efficiencies reported in Tables 12 and 13, respectively.

Table 11: Hardware implementation results for one multiply-accumulate (MAC) unit in 28nm CMOS

| Bits | | | Cyclic act. | Cycle time | Throughput | Cell area | Power |
|---|---|---|---|---|---|---|---|
| Act. | Weight | Acc. | slope $k$ | (ns) | (Gops) | ($\mu m^2$) | (mW) |
| 8 | 8 | 32 | – | 0.31 | 6.5 | 1 298 | 2.78 |
| 3 | 1 | 32 | – | 0.29 | 7.0 | 732 | 1.90 |
| 8 | 8 | 8 | 2 | 0.24 | 8.3 | 521 | 1.60 |
| 8 | 8 | 8 | 4 | 0.25 | 8.1 | 523 | 1.64 |
| 3 | 1 | 8 | 2 | 0.24 | 8.3 | 290 | 0.93 |
| 3 | 1 | 8 | 4 | 0.24 | 8.3 | 285 | 0.87 |

Table 12: Area breakdown of one multiply-accumulate (MAC) unit in 28nm CMOS

| Bits | | | Cyclic act. | Cell area efficiency ($\mu m^2$/Gops) | | | | |
|---|---|---|---|---|---|---|---|---|
| Act. | Weight | Acc. | slope $k$ | Multiplier | Accumulator | Cyclic act. | Auxiliary | Total |
| 8 | 8 | 32 | – | 96 (48%) | 91 (46%) | – | 12 (6%) | 199 |
| 3 | 1 | 32 | – | 3 (2%) | 93 (89%) | – | 9 (9%) | 105 |
| 8 | 8 | 8 | 2 | 31 (49%) | 12 (19%) | 10 (16%) | 10 (16%) | 63 |
| 8 | 8 | 8 | 4 | 33 (50%) | 12 (19%) | 8 (13%) | 12 (18%) | 65 |
| 3 | 1 | 8 | 2 | 2 (5%) | 17 (46%) | 15 (41%) | 3 (8%) | 37 |
| 3 | 1 | 8 | 4 | 2 (5%) | 18 (52%) | 12 (35%) | 3 (8%) | 35 |

## D    USING MORE WEIGHT BITS

Since ARM provides arithmetic operations that handle multiplication between various 8-bit numbers in parallel, we further conduct experiments in which more bits are used for weight quantization.

Table 13: Energy breakdown of one multiply-accumulate (MAC) unit in 28nm CMOS

| Bits | | | Cyclic act. | Energy efficiency (fJ/op) | | | | |
|------|--------|------|---------|------------|-------------|-------------|-----------|-------|
| Act. | Weight | Acc. | slope $k$ | Multiplier | Accumulator | Cyclic act. | Auxiliary | Total |
| 8 | 8 | 32 | – | 144 (34%) | 173 (40%) | – | 111 (26%) | 428 |
| 3 | 1 | 32 | – | 10 (4%) | 197 (73%) | – | 64 (23%) | 271 |
| 8 | 8 | 8 | 2 | 48 (25%) | 29 (15%) | 17 (9%) | 98 (51%) | 192 |
| 8 | 8 | 8 | 4 | 53 (26%) | 28 (14%) | 17 (8%) | 105 (52%) | 203 |
| 3 | 1 | 8 | 2 | 8 (7%) | 42 (37%) | 23 (20%) | 42 (36%) | 115 |
| 3 | 1 | 8 | 4 | 6 (6%) | 42 (40%) | 24 (23%) | 33 (31%) | 105 |

Table 14 displays the classification accuracy, as well as the overflow rate of the final models. Surprisingly, in some cases, we may have a lower overflow rate even when using more bits for the weight quantization. We also collect the accuracy degradation from the full precision network. Our results show that the best performance is achieved when we use 4-bit weights, which is close to the full-precision result (around 0.7% degradation).

Table 14: Results for WrapNet with more bits for weight quantization, where we use ternary weights for 2-bit.

| Bits | Overflow Rate | Accuracy | Degradation |
|------|---------------|----------|-------------|
| 1 | 1.24% | 90.81% | 1.64% |
| 2 | 0.12% | 91.14% | 1.31% |
| 3 | 0.02% | 91.55% | 0.90% |
| 4 | 0.04% | **91.73**% | 0.72% |
| 5 | 0.4% | 91.20% | 1.25% |

