# OpenReview forum: "WrapNet:  Neural Net Inference with Ultra-Low-Precision Arithmetic"
_ICLR.cc/2021/Conference — ICLR 2021 Poster_

### Official Review · AnonReviewer2 · 2020-10-27
**An Empirical Parameter Adjustment Scheme**

**Rating:** 5
**Confidence:** 4

**Review:**

Summary:

The authors propose a scheme named WrapNet to further reduce the bit width of accumulation operations in the deep neural network inference process. Through a novel cyclic activation and regularizer penalizing overflow, the proposed method can be implemented in processors with or without specialized vector instructions using less time, area, and energy cost.


Strength:

--The manuscript is presented well and quite easy to follow.

--The overall idea makes sense and the proposed method can reduce computational resources and speed up computing greatly in the accumulation operations.


Weakness:

--The step-size selection strategy is more like an empirical parameter adjustment, lack of a uniform selection strategy.

-- The results are not so convincing due to the lack of experiments on large-scale models like ResNet50 on ImageNet.


Comments:

(1) The step-size selection strategy seems trying to balance the quantization error and the overflow rate. However, it is more like an empirical adjustment. When the DNN model and the dataset changes or there is a new task, how can it be used in new scenarios because the overflow rate may differ a lot? I think there is a lack of a uniform selection strategy.

(2) The batch normalization layer may change the distributions of weights and activations. In fact, there are some works which have successfully remove batch normalization with comparable performance. And removing batch normalization can better accelerate the DNN computing. In a network without batch normalization, the overflow rate may differ from DNNs with it. Can you do some experiments for comparison and give a detailed explanation of the proposed method used in the DNN model without batch normalization.

(3) The model is pre-trained with quantized weights and full-precision activations. Can the proposed method be applied into the model from scratch given the estimated step size? And how it performs?

(4) It’s better to give the performance of the proposed method on ResNet50, ImageNet to make the results more convincing.

---

> ### Author Response · Authors · 2020-11-23
> **Response to Reviewer 2**
>
> We thank the reviewer for constructive feedbacks and valuable comments.
>
> **(1) The step-size selection strategy is more like an empirical parameter adjustment, lack of a uniform selection strategy.**
>
> The step-size selection is a uniform strategy since for almost all networks or datasets, the quantization step-sizes are estimated from the same pre-defined initial overflow rate (5%). Even for quantization methods such as [1],[2], they achieve the step-size by exponential moving averages or learned by gradient methods which may vary from datasets and networks. On the other hand, for DNNs we usually assume the domain of testing data is the same as (or close to) the domain of training data, and the output distributions from testing and training data should be close, which  leads to a close overflow rate. We would point out that accuracy loss after domain shift is a major problem even for full precision networks, and it is common to use data augmentation at test time, or fine-tuning on a new/updated distribution, to cope with these problems.  Similar domain shift strategies could certainly be deployed here, although this is outside our scope.
>
> **(2) Removing batch normalization (BN) layer.**
>
> We would like to cite the paper if you could give us the exact reference for removing BN with comparable performance. Compared to convolution layers, the overhead of BN layers are small since flops of BN are linear to the input size (the inference time of BN layer is around 1.6% of its corresponding GEMM kernel in average). In addition, BN can be folded into convolution layers during inference. However, we can still apply our method for networks without BN layers, following the exactly same steps as described in Section 3 & 4.  Due to the limited time, we only implement WrapNet for a VGG-7 network without BN on CIFAR10. With ternary weights and an 8-bit accumulator, we have accuracy 83.63%, where a ternary weight and full-precision activation has accuracy 87.61%. The experiment demonstrates that our method also works on networks without BN.
>
> **(3) Can the proposed method be applied into the model from scratch given the estimated step size? And how it performs?**
>
> We train a ternary weight, 8-bit accumulator ResNet18 with the same quantization step-size in the Table 6 from scratch, and we achieve top-1 accuracy 60.01%. Although the accuracy drops around 2%, we remark that training a network with ternary weight and full precision activation from scratch also leads to 1.2% accuracy drop. The result demonstrates that our method can indeed be applied to the network that is trained from scratch.
>
> **(4) It’s better to give the performance of the proposed method on ResNet50, ImageNet to make the results more convincing.**
>
> We provide the TWN and 8-bit accumulator result for ResNet50(top-1 accuracy 71.62%), and we will add the ResNet50 results to Table 6 upon acceptance.
>
> **References:**
>
> [1] Jacob, Benoit, et al. "Quantization and training of neural networks for efficient integer-arithmetic-only inference." Proceedings of the IEEE Conference on Computer Vision and Pattern Recognition. 2018.
>
> [2] Esser, Steven K., et al. "Learned step size quantization." arXiv preprint arXiv:1902.08153 (2019).

---

### Official Review · AnonReviewer1 · 2020-10-27
**Original work that allow efficient low-precision accumulators in inference via a novel cyclic activation function.**

**Rating:** 7
**Confidence:** 3

**Review:**

This paper presents a method (WrapNet) for the problem of efficient low-precision inference. The main contribution is to allow efficient low-bit (e.g., 8 bit) accumulators via the use of a novel cyclic activation function that constrains the value space of the layer outputs, while still allowing good accuracy.

Pros:

The paper is clearly written, although some training details would be nice to see in the main paper instead of the appendix.

Cyclic activation functions have been used in other works, but this work is to my knowledge among the first to use a cyclic ReLU and most likely the first to use it in order to reduce the output value space in order to do more efficient accumulation in inference hardware.

The significance of this work lays in the fact that this is the first paper that allows lower precision accumulators while still demonstrating quite good performance, even in the case of demanding Imagenet classification. The low-power and low-latency inference are of key importance for more widespread use of neural networks in e.g., low-power devices. Actually the performance is almost as good as with other low-precision schemes, which use high-precision accumulators. In addition, the paper has quite extensive study and analysis of performance in various hardware architectures.

Cons:

The method introduces various additional hyperparameters, e.g., regularizer coefficient, transition slope, etc. Although these can be tuned with a validation set.

In results Table 6, why we are not given results that would be comparable to BWN-QA and TWN-QA. For instance, WrapNet with 3 bits/4bits activation and e.g., 12-16 bits accumulator.

Some instability of training seems to happen with different initial overflow rates.

The training setup is relatively complicated  (layer-by-layer training) when using carry variance reduction regularizer.

Typo maybe in page 2 in "reference (fbg, 2018)"

Edit: I am happy with the author responses and have raised the score accordingly to 7.

---

> ### Author Response · Authors · 2020-11-23
> **Response to Reviewer 1**
>
> We thank the reviewer for constructive feedbacks and valuable comments.
>
> **(1) The method introduces various additional hyperparameters, e.g., regularizer coefficient, transition slope, etc. Although these can be tuned with a validation set.**
>
> Yes, we have additional hyperparameters. However, in our experiments, we shared most of the hyperparameters among different network structures and datasets, and all of them achieve comparable results to their 32-bit accumulator counterpart. It will be an interesting future direction to make the hyperparameters learnable.
>
> **(2) In results Table 6, why we are not given results that would be comparable to BWN-QA and TWN-QA. For instance, WrapNet with 3 bits/4bits activation and e.g., 12-16 bits accumulator.**
>
> Thanks for the suggestion, for a ternary weight, 4-bit activation and 12-bit accumulator network, we get 63.73% accuracy, which is comparable to TWN-QA. WrapNet focuses more on the bit-width of the accumulator. Given the target bit-width of the accumulator, we select the quantization step-sizes based on the initial overflow rates, which means when we have a 12-bit accumulator, in general, we could have a larger activation bit-width than 4-bit for a TWN.
>
> **(3) Some instability of training seems to happen with different initial overflow rates.**
>
> Yes, the instability happens when we have large initial overflow rates. However, in Table 3 & 4, we show that within a proper range and with the help of overflow regularizer, the results are stable and comparable.
>
> **(4) The training setup is relatively complicated (layer-by-layer training) when using carry variance reduction regularizer.**
>
> Carry variance reduction is used only for the special case where vector instructions are not available on the CPU. Since there are numerous carry bits generated during bit-packing, training such a network is naturally more complicated. Keep in mind that most modern processor architectures, including x86 and ARM, provide vector instructions and so variance reduction is not needed on these architectures. However, variance reduction provides a mechanism for deploying low-precision networks on simplified hardware that lacks these capabilities.
>
> **(5) Typo maybe in page 2 in "reference (fbg, 2018)"**
>
> Thanks for pointing this out and we will fix that.

---

### Official Review · AnonReviewer4 · 2020-10-28
**An interesting idea on an important issue**

**Rating:** 7
**Confidence:** 5

**Review:**

This paper explores to solve an often ignored issue in quantization: accumulation precision. As the bit-width of input scales down, the area/energy cost of the accumulator starts to dominate.  The cyclic method proposed by the authors at the first glance is not intuitive. However, it's surprising that the surveyed models could be tuned to live with significant overflows---as long as it can be tuned, which is enabled by the "differentiable overflow" brought by the cyclic method.  There are several issues to be addressed before the paper can be accepted:

(1) In the equations of page3,   the boundary in the 2nd line of f(m) has a 'c(zq)',  is that a typo?

(2) Since the paper only covers CIFAR10 and some ImageNet works that are on the easier side to quantize, such as ResNet18 and VGG, the cyclic method could meet its limitation on ResNet50 and Mobilenet. The authors didn't discuss an important concept: accumulation length. As the accumulation length increases, the events of overflow could rise sharply, and the training could fail without room for the cyclic method to optimize the slope k.

(3) Some comments on the relation between the accumulation length and bit-packing would also be helpful. For example, if the accumulator with 8-way bit-packing is working on the same GEMM, the accumulation length would be reduced by 8---that would be desirable. Although, a higher level reduction would be required then.

In general, the paper is well written and brings attention to the important topic of accumulation for reduced precision inference. The paper attempts to solve the overflow problem, although not perfectly, with a differentiable "failure" approach. The paper provides great hardware insights for hardware/software co-design. Therefore, I recommend the paper to be accepted on the condition that the authors could address my comments fairly.

---

> ### Author Response · Authors · 2020-11-23
> **Response to Reviewer 4**
>
> We thank the reviewer for constructive feedbacks and valuable comments.
>
> **(1) In the equations of page3, the boundary in the 2nd line of f(m) has a 'c(zq)', is that a typo?**
>
> Thanks for catching the typo. We will fix that.
>
> **(2) Accumulation length and results for ResNet50**
>
> Thanks for the suggestion. We agree with the reviewer that accumulation length is an important concept in this case, since we will expect more overflows as more elements are added. While we do not name it explicitly, we explain in our paper that the accumulation of many activation-weight products (i.e., a large accumulation length) leads to overflow if low-precision accumulators are used. However, our results show that WrapNet works for several network structures, which suggests that WrapNet covers reasonable and relevant accumulation lengths. To be specific, within the architectures presented, the largest accumulation length was of $3\times 3\times 512=4608$. To further validate WrapNet, we also present the results of our WrapNet on ResNet50 with ternary weights and 8-bit accumulators, in which we obtain a top-1 accuracy of 71.62%.
>
> **(3) Some comments on the relation between the accumulation length and bit-packing.**
>
> We would like to start by mentioning that, if all accumulators share the same bit-width, i.e. 8-bit, then the numerical results for an inner-product (or convolution) will remain the same no matter whether you divide one accumulation into several partial-sums or not, since in modular arithmetic (A + B) mod C = (A mod C + B mod C) mod C.
>
> The attractiveness of dividing one inner-product into several partial-sums is that each partial-sum is represented with fewer bits than the final result. Nonetheless, as these partial-sums are added together, they will ultimately need as many bits as the final result to be represented; otherwise, overflow might occur. So for example, let’s say we want to add 16 8-bit numbers. This means that the final result needs 12 bits to be represented. We can start by adding 8 8-bit numbers, which results in an 11-bit number. The fact that now we have an 11-bit number makes bit-packing not as efficient as before, as these operations are offered for 8- or 16-bit numbers. In this example, we would need to start using 16-bit operations to add the two 11-bit numbers and get the final 12-bit result. One of the advantages of WrapNet is that all these partial-sums will be represented with 8-bit numbers, so bit packing can be performed using the more efficient 8-bit instructions only, regardless of the accumulation length.
>
> Once we use WrapNet, the reviewer is right that if 8-way bit-packing is working on the same inner-product, then the number of vector-add (VADD) instructions needed would be reduced by 8. However, as the reviewer mentions as well, one would need additional VADD instructions to accumulate the 8 individual partial-sums. Then, depending on how many inner-products need to be computed, the approach suggested by the reviewer might not offer any gains. For example, consider that you need to perform 8 inner-products with accumulation length of 256 each. One approach would be to use each way of bit-packing for one inner-product. Such an approach would require 255 VADD instructions to complete the operation. Another approach would be to compute one inner-product at a time by using all 8 ways of bit packing. In this case, one inner-product (without the final reduction) will take 31 VADD instructions. After doing this 8 times, we end with eight 8-entry partial-sum vectors to be added together, which needs 7 instructions, for a total of $31\times 8+7=255$ VADD instructions. Hence, the second approach did not reduce the number of VADD instructions, and what is more, it will need additional instructions to rearrange the partial-sums to be reduced together so that they are on different memory words (as required by the VADD instruction).
>
> While there are many different ways in which the accumulation could be implemented, the key point of this paper is to show that accumulation can be performed with low-resolution registers, even amidst the occurrence of overflows. To illustrate the advantages of this approach, we have implemented reference hardware and software designs, but we do not intend to claim that they are the only ways of implementing WrapNet. For example, in hardware, WrapNet can also be implemented using spatial architectures, such as systolic arrays, which use several adders per inner-product -- the complexity of these adders would be reduced as well thanks to WrapNet. Nonetheless, the study of other software and hardware implementations that use WrapNet (and the quantification of their gains) is a complex topic that is left for future work.

---

### Official Review · AnonReviewer3 · 2020-10-28
**Good paper; however overheads regarding $f$ needs to be quantified**

**Rating:** 7
**Confidence:** 4

**Review:**

## Summary:

The paper introduces a new neural network layer that enables training NNs with quantized activations using reduced bit-width accumulators. The cyclic activation layer makes overflows smooth, instead of being discontinuous and this enables achieving better accuracy on quantized networks on reduced bit-width accumulators. They also introduce overflow and carry penalties to dissuade the training regime from reaching overflow states.

## Strengths:
* New NN layer (operator), cyclic activation layer that makes quantized accumulated outputs smooth when overflow is present. The authors show that even at large overflow rates, the network accuracy does not degrade that much.
* Introduction of the overflow penalty to dissuade overflow and together with the cyclic activation layer achieves higher accuracy even at high overflow probabilities.
* Evaluation using both vector instructions and carry penalty when vector instructions are not present; they show that carry penalty mitigates carry contamination leading to a closer accuracy compared to their vector implementation.
* Hardware analysis of possible cost savings for reduced accumulator implementations in a MAC unit.

## Weaknesses and Questions for authors:

* Intuitive explanation as to how $f$ was designed is missing.
* Isn’t computing $f$ need to be done in full precision or in higher precision than the accumulator? For instance say $k = 8$, then computing $k \times 2^{b-1}$ needs $b+2$ bits to compute. What is the overhead of computing $f$ in higher precision? Based on the results from section 4.5, this overhead may not be that much, but still needs to be quantified.
* Also, the overhead of computing $f$ varies with $k$. The same goes for hardware analysis. A quantification is needed to understand the tradeoffs.
* I assume that loss function with regularizers is computed in full precision. Correct me if I am wrong.

---

> ### Author Response · Authors · 2020-11-23
> **Response to Reviewer 3**
>
> We thank the reviewer for constructive feedbacks and valuable comments.
>
> **(1) Intuition for the design of cyclic activation**
>
> The intuition of f is that besides the cyclic property, we would like to make the function “smooth” and continuous, or otherwise it will be hard to train (see Table 2.). As a result, we would like to have a transition slope to control the “smoothness” of the cyclic activation.
>
> **(2) Overhead for f during inference**
>
> Thanks for the suggestions, we will add the results and discussion for both software and hardware costs in the camera-ready version. We start by noting that the cyclic activation f is inserted after the convolution block where overflow happens naturally. Furthermore, the input and output of f lay in the range of the accumulator’s bit-width. For example, to compute $k2^{b-1} - km$, we could calculate it as $k(2^{b-1} - m)$. Since in this case, $m > (k/(k+1))2^{b-1}$, then we have that $2^{b-1} - m < (1/(k+1))2^{b-1}$, which can be represented with b bits of resolution. Thus, we can compute the output of the cyclic activation without requiring additional bits of resolution.
>
> In terms of implementation complexity, we note that, for a fixed k and b, the cyclic activation f is performing a multiplication and addition with constants, and a selection, all of which are hardware-friendly operations. We emphasize that the complexity of a multiplication (addition) with a constant is significantly lower than the complexity of a general, two-input multiplication (addition). Hence, we expect the cyclic activation to have a low implementation overhead.
>
> To quantify this overhead, we have implemented the cyclic activation for an accumulator bit-width b=8 and for different values of k in 28nm CMOS technology. We observed that the cyclic activation’s area efficiency is between 68% and 88% that of an 8-bit accumulator, while its energy efficiency is between 55% and 63%. In other words, the cyclic activation has an area comparable to that of an accumulator, but lower power consumption. Once we include the cyclic activation overhead in our hardware results, we observe that WrapNet (with an 8-bit accumulator) still improves area- and energy-efficiency up to a factor of 3.2x and 2.6x, respectively, when compared to a traditional MAC unit (32-bit accumulator) without cyclic activation. This demonstrates that our WrapNet approach still offers savings in hardware even when considering the overhead from the cyclic activation.
>
> Nonetheless, we would like to add that the overhead imposed by the cyclic activation can be reduced by using spatial architectures, such as systolic arrays, to perform matrix multiplication. In such spatial architectures, several multipliers and adders are used to compute one inner-product. Since we only need one cyclic activation for each inner-product, the overhead of the cyclic activation will be amortized by the area and power of several multipliers and adders contributing to the same inner-product. We would also like to add that WrapNet also reduces the complexity of the adders used in a spatial architecture.
>
> Regarding the impact of k on the hardware implementation of the cyclic activation, we do not observe a clear trend on the power consumption with respect to k. However, we have observed that the area reduces as k increases. Intuitively this makes sense, as when k increases, there are fewer numbers that deviate from the identity function (if $k=\infty$, then the cyclic activation would have no overhead). However, as mentioned before, the area and power of the cyclic activation are never worse than those of an accumulator with the same bit-width, regardless of k.
>
> We also provide the overhead of the cyclic function in software. Compared to the cost of corresponding GEMM kernel, the overhead of the cyclic activation is relatively small. Following is the speed test for our GEMM kernel and cyclic activation:
>
> | Input size | GEMM  | Cyclic |
> |------------|-------|--------|
> | 64x56x56   | 3.467 | 0.708  |
> | 128x28x28  | 2.956 | 0.370  |
> | 256x14x14  | 2.499 | 0.169  |
> | 512x7x7    | 2.710 | 0.087  |
>
> **(3) Training in full precision**
>
> Yes, you are right. The training is based on full precision.

---

### Author Response · Authors · 2020-11-23
**Updates**

We thank all the reviewers for carefully examining our paper.  We provide the requested result of our WrapNet on ImageNet with architecture ResNet50. Due to the limited compute time, we only provide a ternary weight ResNet50 with 8-bit accumulators, where the activation is within 4-bit. We achieve top-1 accuracy 71.62%, where the TWN-QA with 32-bit accumulator has accuracy 72.50%. We will add the full results of ResNet50 into the main table upon acceptance.

---

### Decision · Program_Chairs · 2021-01-07
**Final Decision**

**Decision:**

Accept (Poster)

**Comment:**

Most of the reviewers agree that this paper presents an interesting idea. Practically implementing a BNN that gains real world speedup is challenging, and as past work [1] showed, the bottleneck could shift into other layers(besides the accumulation). The paper would benefit from a thorough discussion about the practical impact in implementing the proposed method and relation to past work.

The meta-reviewer decided to accept the paper given the positive aspects, and encourages the author to further improve the paper per review comments.

Thank you for submitting the paper to ICLR.

[1] Riptide: Fast End-to-End Binarized Neural Networks